# DNA Methylation of Window of Implantation Genes in Cervical Secretions Predicts Ongoing Pregnancy in Infertility Treatment

**DOI:** 10.3390/ijms24065598

**Published:** 2023-03-15

**Authors:** Quang Anh Do, Po-Hsuan Su, Chien-Wen Chen, Hui-Chen Wang, Yi-Xuan Lee, Yu-Chun Weng, Lin-Yu Chen, Yueh-Han Hsu, Hung-Cheng Lai

**Affiliations:** 1International Ph.D. Program for Cell Therapy and Regeneration Medicine, College of Medicine, Taipei Medical University, Taipei 110301, Taiwan; 2Department of Obstetrics and Gynecology, School of Medicine, College of Medicine, Taipei Medical University, Taipei 110301, Taiwan; 3Department of Obstetrics and Gynecology, Hai Phong University of Medicine and Pharmacy, Hai Phong 04254, Vietnam; 4Department of Obstetrics and Gynecology, Shuang Ho Hospital, Taipei Medical University, New Taipei 235041, Taiwan; 5Translational Epigenetics Center, Shuang Ho Hospital, Taipei Medical University, New Taipei 235041, Taiwan; 6Dr. Wang Reproductive Fertility Center, Taipei 110007, Taiwan; 7Graduate Institute of Clinical Medicine, Taipei Medical University, Taipei 110301, Taiwan; 8Taipei Fertility Center, Taipei 110016, Taiwan

**Keywords:** cervical secretion, DNA methylation, noninvasive, window of implantation, endometrial receptivity, in vitro fertilization, machine learning, boruta algorithm

## Abstract

Window of implantation (WOI) genes have been comprehensively identified at the single cell level. DNA methylation changes in cervical secretions are associated with in vitro fertilization embryo transfer (IVF-ET) outcomes. Using a machine learning (ML) approach, we aimed to determine which methylation changes in WOI genes from cervical secretions best predict ongoing pregnancy during embryo transfer. A total of 2708 promoter probes were extracted from mid-secretory phase cervical secretion methylomic profiles for 158 WOI genes, and 152 differentially methylated probes (DMPs) were selected. Fifteen DMPs in 14 genes (BMP2, CTSA, DEFB1, GRN, MTF1, SERPINE1, SERPINE2, SFRP1, STAT3, TAGLN2, TCF4, THBS1, ZBTB20, ZNF292) were identified as the most relevant to ongoing pregnancy status. These 15 DMPs yielded accuracy rates of 83.53%, 85.26%, 85.78%, and 76.44%, and areas under the receiver operating characteristic curves (AUCs) of 0.90, 0.91, 0.89, and 0.86 for prediction by random forest (RF), naïve Bayes (NB), support vector machine (SVM), and k-nearest neighbors (KNN), respectively. SERPINE1, SERPINE2, and TAGLN2 maintained their methylation difference trends in an independent set of cervical secretion samples, resulting in accuracy rates of 71.46%, 80.06%, 80.72%, and 80.68%, and AUCs of 0.79, 0.84, 0.83, and 0.82 for prediction by RF, NB, SVM, and KNN, respectively. Our findings demonstrate that methylation changes in WOI genes detected noninvasively from cervical secretions are potential markers for predicting IVF-ET outcomes. Further studies of cervical secretion of DNA methylation markers may provide a novel approach for precision embryo transfer.

## 1. Introduction

Infertility affects about 20% of couples worldwide, a rate that is currently increasing [1]. Consequently, the use of assisted reproductive technologies (ARTs) is also increasing substantially, by up to 5–10% annually [2]. The number of ART cycles conducted annually has reached 3.3 million [3]. Although in vitro fertilization (IVF) is the most effective ART approach, its resultant pregnancy rates remain unsatisfactory.

Implantation outcomes in IVF largely depends on the embryo, endometrium, and embryo-endometrial synchrony. Thanks to the application of time-lapse-based culture systems and preimplantation genetic testing for aneuploidy, euploidy embryos can now be selected for transfer. Nevertheless, the ongoing pregnancy rate from a single euploid embryo transfer cycle is less than 60% [4]. This evidence highlights the unmet need for assessment of endometrial quality before embryo transfer to improve outcomes. To date, the endometrium has been considered a ‘black box’ in regard to its involvement in the implantation process. The endometrium becomes optimally receptive to supporting embryo implantation during a limited period, termed the ‘window of receptivity’ or ‘window of implantation’ (WOI), which normally lasts from days 4–10 of progesterone production or administration [5,6,7]. Multiple investigators have attempted to identify the transcriptomic pattern denoting a receptive WOI status compared with other menstrual phases. However, significant discrepancies have been observed in the genes involved in the WOI, and its comprehensive transcriptomic pattern has yet to be elucidated because of differences in endometrial biopsy timing and mRNA profiling methods [8].

Recently, endometrial transcriptomic profiles across menstrual cycles were revealed by single-cell RNA sequencing (scRNA-seq). scRNA-seq enables the definition of the global gene expression profiles of single cells and the dissection of cell population heterogeneity that was previously hidden. The breakthroughs have provided a normative baseline for further investigations of endometrial biology and more precise identification of WOI-associated genes from endometrial cells [9]. Further investigation of the regulation of these genes may shed light on the molecular changes that affect whether embryo transfer cycles result in pregnancy.

DNA methylation, a key regulator of gene expression, is involved in controlling endometrial functions, and plays important roles in endometrial regeneration and transformation via regulation of gene expression during progenitor cell proliferation and differentiation [10]. Changes in the hormonal milieu and environmental conditions during the menstrual cycle may alter the endometrial DNA methylome, affecting endometrial functions across the menstrual cycle, including its receptivity to supporting embryo implantation [10,11]. Evidence from human specimen and animal studies demonstrates that altered methylation statuses in key genes and global DNA methylome changes are associated with impaired endometrial receptivity and implantation failure [11,12,13,14,15,16,17,18,19,20,21,22]. It is thus anticipated that the DNA methylation status of WOI genes is associated with the implantation outcome in embryo transfer cycles.

Cervicovaginal materials have been used as a noninvasive surrogate in investigations of endometrial conditions. Metabolomic, protein, and cytological markers from cervicovaginal fluids have been studied to identify endometrial conditions [23,24,25,26]. In assisted reproduction, while markers from cervicovaginal washing or lavages, including glycodelin A and cytokines, have been associated with implantation outcome, they have not yet been successfully validated [27,28,29,30]. Interestingly, it is evident that DNA methylation markers from cervicovaginal materials can be used to detect endometrial cancer with high accuracy [31,32,33]. Furthermore, our previous study showed that DNA methylation profiles of mid-secretory cervical secretions are associated with IVF-ET outcome [34]. Herein, we hypothesized that the DNA methylation status of WOI genes from cervical secretions can be used to predict ongoing pregnancy in embryo transfer cycles. Machine learning (ML) is a valuable tool that can complement conventional statistics to identify relevant biomarkers from high dimensional omics datasets for diagnostic or predictive tasks [35,36]. Thus, we used a supervised ML approach to investigate methylation changes in WOI genes from noninvasively collected mid-secretory phase cervical secretions to test the potentials of differentially methylated genes to predict ongoing pregnancy in embryo transfer cycles.

## 2. Results

### 2.1. Methylomic Profiles of Mid-Secretory Phase Cervical Secretions

No statistically significant differences in important demographic and clinical characteristics were found between the pregnancy and nonpregnancy groups in either the discovery or verification set (Table 1).

Next, we tested the methylation status of the selected WOI genes from the mid-secretory phase cervical secretion samples (n = 31 pregnancy, n = 37 nonpregnancy) in the array sample set. After quality control and normalization, 743,730 probes remained. Methylomic profiles were then used to extract WOI gene promoter probes for further analysis (Figure 1). The distribution of methylation values for each probe are demonstrated in a density plot (Appendix A).

### 2.2. Endometrial WOI Genes and Their Promoter Probes

One hundred and fifty-eight WOI genes were selected from the endometrial single-cell transcriptomic profiles. These genes belong to three groups, including genes whose expressions determine the endometrial cycle phase and subphase (i.e., top phase and subphase-defining genes), transcriptional factor coding genes, and secretory protein coding genes associated with the entrance and/or exit of the WOI (Appendix A). A total of 2708 promoter probes of the 158 WOI genes were extracted from the mid-secretory cervical secretion methylomic profiles (Appendix A).

### 2.3. Selection of Differentially Methylated Probes for Predicting Ongoing Pregnancy

From 2708 promoter probes in 158 WOI genes, 152 probes in 87 genes were found to have significant differences in methylation between the pregnancy and nonpregnancy sample groups (Appendix A, Figure 2). The best model from the four ML algorithms (RF, NB, SVM, and KNN) using these 152 DMPs for predicting ongoing pregnancy in the array dataset achieved accuracy rates of 80.65%, 83.44%, 83.47%, and 69.03%, respectively (Appendix A), and AUC values of 0.85, 0.87, 0.88, and 0.79, respectively (Appendix A).

All DMPs were ranked by importance level compared with the randomly generated shadow attributes via BA. The results of the variable selection process were demonstrated by variable important ranking and classifier run history before (Appendix A) and after the tentative fix (Appendix A, Figure 3a, Appendix A). The process identified 15 DMPs in 14 genes as the most statistically relevant to ongoing pregnancy status (Appendix A, Figure 3a,b).

The best model from each of the four ML algorithms (RF, NB, SVM, and KNN) using these 15 DMPs for predicting ongoing pregnancy in the array dataset yielded accuracy rates of 83.53%, 85.26%, 85.78%, and 76.44%, respectively, and AUC values of 0.90, 0.91, 0.89, and 0.86, respectively (Table 2, Figure 3c).

### 2.4. Verification of Three Selected Candidate Genes for Predicting Pregnancy Outcome

Three selected genes, SERPINE1, SERPINE2, and TAGLN2, were verified using the independent set (qMSP sample set) of 30 pregnancy and 35 nonpregnancy cervical secretion samples.

In the array sample set, the methylation level (β-value) of each gene promoter probe differed significantly between the two sample groups. SERPINE1 (cg19722814) and TAGLN2 (cg03138854) had higher methylation levels, whereas SERPINE2 (cg19371203) had a lower methylation level in pregnancy compared with nonpregnancy samples (Figure 4a). Predicting ongoing pregnancy in the array set using the DMP set with the best models via RF, NB, SVM, and KNN achieved accuracy rates of 73.15%, 74.32%, 73.68%, and 75.62%, respectively, and AUC values of 0.82, 0.84, 0.84, and 0.82, respectively (Appendix A).

In the qMSP sample set, the between-sample group methylation level difference trend in each gene was similar to that in the array sample set, and the differences remained statistically significant for SERPINE2 and TAGLN2 (Figure 4b). The best model from RF, NB, SVM, and KNN using the three-gene combination achieved accuracy rates of 71.46%, 80.06%, 80.72%, and 80.68%, respectively, and AUC values of 0.79, 0.84, 0.83, and 0.82, respectively (Table 3, Figure 4c).

## 3. Discussion

To the best of our knowledge, this is the first study to report that the methylation status of single cell level-identified WOI genes from cervical secretions differ between IVF cycles that lead to pregnancy versus nonpregnancy. These data show that ML algorithms using differentially methylated WOI genes can accurately predict ongoing pregnancy. Our findings support further studies using DNA methylation markers from cervical secretions as a noninvasive approach for identifying receptive cycles to potentially determine precision embryo transfer and help improve IVF success.

Previously, using a genome-wide analysis of methylation profiles from cervical secretions in embryo transfer cycles, we showed that unsupervised clustering using 2000 DMPs could correctly classify 86% of samples [34]. Consistent with that evidence, the results herein support the conclusion that DNA methylation changes detected from cervical secretions are highly relevant to the IVF-ET outcome. A total of 152 DMPs in 87 genes were found to have significantly different methylation levels between pregnancy and nonpregnancy samples. Using these DMPs with the four ML algorithms predicted ongoing pregnancy with accuracy rates from 69.03% to 83.47%, and AUC values from 0.79 to 0.88. Further refinement showed that 15 DMPs in 14 genes were most relevant to ongoing pregnancy status. Furthermore, the four ML algorithms based on these DMPs achieved good-to-excellent prediction performance, with AUC values from 0.86 to 0.91 and accuracy rates from 76.44% to 85.78%.

Endometrial DNA methylation plays an important role in embryo implantation. Methylation changes in genes involved in inflammatory response regulation, adhesion molecule synthesis, transcriptional factors, and tissue remodeling in the endometrium have been associated with poor implantation outcomes in IVF treatment [11,14,15,21,22]. Herein, we utilized endometrial WOI genes, which have been comprehensively identified at the single cell level, to investigate methylation changes using cervical secretions. Among 14 WOI genes, the methylation statuses of which were identified as most relevant to ongoing pregnancy status, nine (MTF1, ZNF292, ZBTB20, TCF4, CTSA, DEFB1, GRN, SERPINE1, and THBS1) were found in unciliated endometrial epithelial cells, and the other five (BMP2, SFRP1, STAT3, SERPINE2, and TAGLN2) in stromal fibroblasts. Expression levels of these genes differ between the WOI and other menstrual cycle phases or subphases [9].

Although the biological functions of these 14 candidate WOI genes in the endometrium are not fully understood, eight were previously shown to be involved in implantation and decidualization. THBS1 encodes for thrombospondin-1, which is involved in tissue remodeling [37]. In humans, THBS1 is upregulated in receptive compared with pre-receptive endometria [38], and downregulated in patients with recurrent implantation failure [39]. Secreted frizzled-related protein 1 (SFRP1) is a modulator/inhibitor of Wnt/Fz sFRP1, which inhibits cell proliferation. SFRP1 is more abundant in stromal cells, and its expression is higher during the proliferative compared with the secretory phase [40]. The gene was upregulated in clomiphene citrate treatment cycles, and may be the major cause of thin endometrium [41]. Bone morphogenetic proteins (BMPs) are multi-functional growth factors that belong to the transforming growth factor beta superfamily [42]. BMP2 regulates endometrial remodeling during the decidualization process via different signaling pathways (e.g., ALK3-SMAD1/5, cAMP/PKA/Wnt4, Wnt4/β-catenin [43,44,45,46,47]), and its expression may be influenced by progesterone [48]. TCF4 and STAT3 encode for transcriptional factors and are involved in cell growth and differentiation in different tissues, including human endometrium. Impaired STAT3 expression is a possible cause of altered endometrial receptivity and decidualization via different pathways [49,50,51,52,53,54]. Both STAT3 and TCF4 are downregulated in the decidua of patients with recurrent spontaneous abortion compared with those with normal pregnancy [55]. In a mouse model, an impaired LIF/STAT3 pathway resulted in the failure of decidualization and embryo implantation [56]. SERPINE1, SERPINE2, and TAGLN2 are secretory protein coding genes. SERPINE1 encodes for the plasminogen activator inhibitor-1 protein, which is well known for its roles in stromal cell remodeling in different tissues [57]. In the endometrium, its expression in decidual cells is influenced by steroid hormones [58,59,60]. This protein, in turn, regulates the proteolytic degradation of the extracellular matrix and fibrinolysis in endometrial cells, which are essential for decidualization and the invasion of the trophoblast during implantation [61,62]. In addition, polymorphisms of the gene have been associated with implantation failure [63,64,65]. SERPINE2, also known as protease nexin-1, belongs to the serine protease inhibitor (SERPIN) superfamily. SERPINE2 is the most upregulated polysaccharide intercellular adhesin in the mouse endometrial epithelium [66] during estrous and in human mid-secretory phase endometrium [67], supporting its role in endometrial remodeling during the WOI. TAGLN2 is upregulated in the receptive, compared with the pre-receptive, endometrium [68], and is involved in regulating cell invasion, migration, and differentiation [69]. In a rabbit model, increased TAGLN2 expression at implantation sites in early pregnancy was under the influence of live embryo(s) and elevated steroid hormones [70]. Cumulatively, these genes have the potential to serve as key regulators of embryo implantation, so that methylation alterations may dysregulate their functions during the WOI and affect implantation outcomes. The roles of three other transcriptional factors (ZNF292, ZBTB20, and MTF1) and three other secretory proteins (CTSA, DEFB1, and GRN) in implantation have not yet been elucidated and thus require further investigation.

Consistent with our previous study that supported the use of cervical secretion as a proxy for assessing endometrial receptivity based on pregnancy status [34], we have here provided additional evidence supporting this noninvasive approach by using differentially methylated WOI genes to predict ongoing pregnancy. Although the noninvasive or minimally invasive assessment of endometrial receptivity has been demonstrated using molecular markers, including transcriptomics, epigenomics, proteomics, and metabolomics from endometrial fluids, each has limited performance and needs further validation [71,72]. Moreover, previous attempts using clinical features in ML models to predict IVF outcome have demonstrated modest performance [73,74,75]. Herein, four ML algorithms using 15 DMPs in 14 genes demonstrated the potential for predicting ongoing pregnancy. In addition, methylation levels of three genes (SERPINE1, SERPINE2, and TAGLN2) were verified in an independent sample set that showed consistent methylation difference trends between pregnancy and nonpregnancy samples. Four ML algorithms using three genes have also achieved promising performance in sample classification. These results reaffirm that methylation changes in WOI genes from cervical secretions are associated with endometrial status and have potential as markers for predicting IVF-ET outcomes.

This study had some limitations. First, the sample sizes were relatively small in both the discovery and verification sets. However, the predictive performance was generated from the repeated cross-validation procedure that reduced overfitting and showed the potential for generalizing the proposed algorithms using these DNA methylation markers to other populations. Although the best model from each ML algorithm using different combinations of the candidate methylation markers achieved promising performance, the utility of these models needs further evaluation. Studies with larger samples would help validate the predictive performance of these candidate genes. In addition, using the candidate genes in combination with important clinical factors such as women’s age and embryo quality may result in more precise prediction. Second, we used only morphological criteria to assess embryo quality. Implantation failure may have resulted from abnormal embryos, which may have caused some misinterpretation in these findings. Testing these DNA methylation markers in only those cycles when euploid embryos are transferred would rule out pregnancy failure from impaired quality embryos and further confirm the predictive performance of their differentially methylated status. Third, cervical secretions were collected immediately before embryo transfer, which is unsuitable for clinical use. Nevertheless, it is worth further studying these candidate genes using late proliferative phase cervical secretions to investigate how their methylation status differs between pregnancy and nonpregnancy cycles, and whether they can be used to predict embryo transfer outcomes because it was evident that the endometrial DNA methylome remains stable during the menstrual cycle up to the late-secretory phase [76]. An investigation with a larger sample size is now planned to validate these methylation markers using late proliferative phase cervical secretions. The confirmation of optimal DNA methylation markers and the development of predictive models are expected to provide timely information on the probability of achieving pregnancy, allowing clinicians and patients to make more well-informed decisions about embryo transfer.

## 4. Materials and Methods

### 4.1. Study Participants and Clinical Samples

This study was approved by the joint institutional review board of Taipei Medical University (TMU-JIRB number: N201703072). Written informed consent was obtained from all participants with the approval of the Ethics Committee.

The participants in the present study were women undergoing embryo transfer during IVF. Cervical secretion samples were collected from eligible participants at multiple IVF centers between August 2018 and May 2020. Standard treatment protocols were used at all IVF centers. Participants had no contraindications for IVF, endometrial thickness greater than 7 mm on the embryo transfer day, and at least one fresh or frozen embryo scheduled for transfer. Embryo quality was assessed according to the Gardner and Schoolcraft grading system [77], as described in our previous study [34]. The participants’ clinical characteristics were recorded. Both fresh and frozen embryo transfer cycles were eligible for this study. Fresh embryos were transferred after IVF/ICSI following ovarian stimulation and oocyte retrieval. In frozen embryo cycles, both natural endometrial cycles and cycles with artificial endometrial preparation using hormone replacement were employed. Women with endometriosis or adenomyosis were pretreated with gonadotropin-releasing hormone for at least 1 month for pituitary suppression. We collected one cervical secretion sample from each participant during her embryo transfer, immediately before the transfer catheter was inserted into the cervical canal, as described in our previous study [34]. After embryo transfer, the women were followed for pregnancy assessment. Cervical secretion samples were classified into the pregnancy or nonpregnancy groups according to the presence of at least one viable intrauterine fetus at 12 weeks of gestation, as described previously [34]. Herein, we used two independent cervical secretion sample sets: one to generate the methylation profiles by using an Illumina 850 K BeadChip array (i.e., array sample set or discovery set), which was then used to extract methylation data on WOI genes, and another to verify the methylation levels of selected candidate genes by quantitative methylation-specific polymerase chain reaction (i.e., qMSP sample set or verification set).

### 4.2. DNA Preparation

The collection of cervical secretions and DNA extraction were performed as described in our previous study [34]. Briefly, cervical secretions were collected using a cotton wool ball, which was placed in a 50 mL centrifuge tube and stored at 4 °C. The cotton wool ball was rinsed with 1 mL phosphate-buffered saline. This solution was then centrifuged at 1000× *g* for 10 min to collect the flowthrough. Genomic DNA was extracted from the flowthrough using the QIAamp DNA Mini Kit (QIAGEN, Hilden, Germany) and bisulfite-converted using the EZ DNA Methylation Kit (Zymo Research Corp., Irvine, CA, USA), in accordance with the manufacturers’ protocols.

### 4.3. Extraction of WOI Genes’ Promoter Probes from Cervical Secretion Methylomics Profiles

Methylomic profiles were generated from the samples in the discovery set using the Infinium MethylationEPIC BeadChip array (Illumina, San Diego, CA, USA), which measures the methylation levels of around 850,000 CpG sites according to Illumina’s standard protocol. In the system, the methylation level of each probe was represented by a β-value. Quality control and normalization for methylation levels derived from type I and type II probes were performed using the beta-mixture quantile (BMIQ) method in the chip analysis methylation pipeline package of R [78], as described previously [34]. Methylation profiles, after normalization and removing probes with single-nucleotide polymorphism, were used for data extraction.

We selected WOI associated genes which were identified from the single-cell transcriptomic profiles, which were previously described [9]. Methylation values (β-values) of promoter probes in each WOI gene were extracted from the mid-secretory phase cervical secretion methylomic profiles.

### 4.4. Measurement of Methylation Levels by Quantitative Methylation-Specific Polymerase Chain Reaction

A panel of selected genes with different importance levels ranked by BA algorithm was used for verification. We measured the methylation levels of these genes with qMSP in the independent set of cervical secretion samples (qMSP sample set) following the protocol described in our previously published studies. Briefly, we designed the primers (Appendix A) using Oligo 7.0 Primer Analysis software (Molecular Biology Insights, Inc., Colorado Springs, CO, USA). qMSP assays were performed on the LightCycler 480 System (Roche, Indianapolis, IN, USA) [31]. Each gene was tested twice in all samples. We normalized the total input DNA amount in each qMSP reaction using the unmethylated type II collagen gene (COL2A1) as a reference. DNA methylation levels were estimated by the difference in crossing point (ΔCp) values, defined by the following formula: Cp of target gene − Cp of COL2A1 [79]. Samples with test results showing a Cp value of COL2A1 > 36 for each gene were defined as the absence of the template DNA [33]. Herein, the relative methylation levels measured by qMSP are presented as 2^−(ΔCp)^.

### 4.5. Statistical and Machine Learning Analyses

#### 4.5.1. Datasets

Array (discovery) dataset

The array dataset consisted of all WOI gene promoter probes extracted from methylomic profiles, which were generated from the array set of cervical secretion samples (array set). The dataset was used to select variables and train and evaluate the four ML algorithms using different sets of differentially methylated probes (DMPs) for predicting ongoing pregnancy.

qMSP (verification) dataset

The qMSP dataset consisted of the selected gene panel, the methylation levels of which were verified by qMSP in the independent set of mid-secretory phase cervical secretion samples (qMSP set). This dataset was used to train and evaluate the performance of the ML algorithms using the gene panel for predicting ongoing pregnancy.

In both datasets, methylation levels of the WOI gene promoter probes were numerically categorized. The binary outcome variable reflected ongoing pregnancy status after embryo transfer, as described above. The positive class in predictive models was defined as “pregnancy” and the negative class as “nonpregnancy”. Clinical characteristics of the samples were obtained and processed as continuous or categorical variables, as appropriate.

#### 4.5.2. Statistical Analyses

Statistical and ML analyses were conducted, and plots created, using SPSS Version 26.0 (IBM Corp., Armonk, NY, USA), and different packages implemented in R (version 4.1.1) [80]. Bivariate analyses were performed for all variables, including clinical characteristics and promoter probes, to test for differences between pregnancy and nonpregnancy sample groups. The chi-squared test or Fisher’s exact probability test were applied to categorical variables, and the nonparametric Mann–Whitney *U* test was applied to continuous variables. A *p*-value < 0.05 was considered statistically significant. We used the R pheatmap package to generate a heatmap to visualize the probes with methylation levels that differed significantly between the two sample groups [81].

#### 4.5.3. Variable Selection

Variable selection is an important ML step to identify the most relevant set of variables for outcome prediction (herein, we use the terms ‘variable’, ‘feature’, and ‘attribute’ interchangeably). First, bivariate analyses were conducted to identify variables that differed significantly between the sample groups; these were used for further refinement. Next, the Boruta algorithm (BA), a tree-based wrapper feature selection algorithm, was used to select the variables most relevant to ongoing pregnancy status. Briefly, BA calculates importance scores for each attribute in the algorithm and compares these with scores for randomly generated shadow attributes. The attributes with importance levels denoted by Z-scores that are higher than those of all random shadow attributes are deemed ‘important’. The shadow attributes are then removed from the process, and the procedure is repeated until the importance assessment is completed for all attributes, or the algorithm is stopped based on a predefined number of runs [82]. Herein, we set a limit of 500 BA runs. In addition, a tentative rough fix step was applied in the BA procedure to make a final decision on variables that remained inconclusive (tentative variables) after all initial iterations. The process was performed using the “Boruta” package in R [82].

#### 4.5.4. Model Train and Evaluation

In model training and evaluation, the binary outcome variable was defined as a 2-level factor, while numerical variables were centered and scaled using the preProcess function in the Caret package in R [83]. Different combinations of selected variables were then used as predictors in four ML algorithms (random forest [RF], naïve Bayes [NB], support vector machine [SVM], and k-nearest neighbors [KNN]) to predict ongoing pregnancy. One hundred repetitions of a fivefold cross-validation were conducted to create 500 held-out test sets used to select the best hyperparameter for each algorithm and assess model performance using the Caret package in R [83]. Hyperparameter tuning in the repeated cross-validation procedure was performed using the default search in the “train” function in the Caret package to select the best model for each algorithm. Predictive performance metrics, including accuracy, positive predictive value, negative predictive value, sensitivity, and specificity, were calculated for the final best model using each algorithm with 500 resamples using the confusionMatrix function in the Caret package in R. Receiver operating characteristic (ROC) curves were constructed and area under the ROC curve (AUC) values were calculated from sensitivity and specificity values for each threshold from the repeated cross-validation procedure using the pROC package in R [84]. We focused on accuracy and AUC as the main metrics of predictive performance of the four ML models.

## 5. Conclusions

The findings of this study suggest that methylation changes in WOI genes, detected noninvasively from cervical secretions, can be used to predict ongoing pregnancy in embryo transfer cycles. Methylation changes in these critical genes may dysregulate endometrial receptivity during the WOI, impacting embryo transfer outcomes. Future studies using DNA methylation markers detected from late proliferative cervical secretions may provide a novel noninvasive approach for precisely predicting pregnancy ahead of embryo transfer. Such timely information provided by DNA methylation marker-based predictive models would be clinically valuable for precision embryo transfer and improving IVF outcomes.

## Figures and Tables

**Figure 1 ijms-24-05598-f001:**
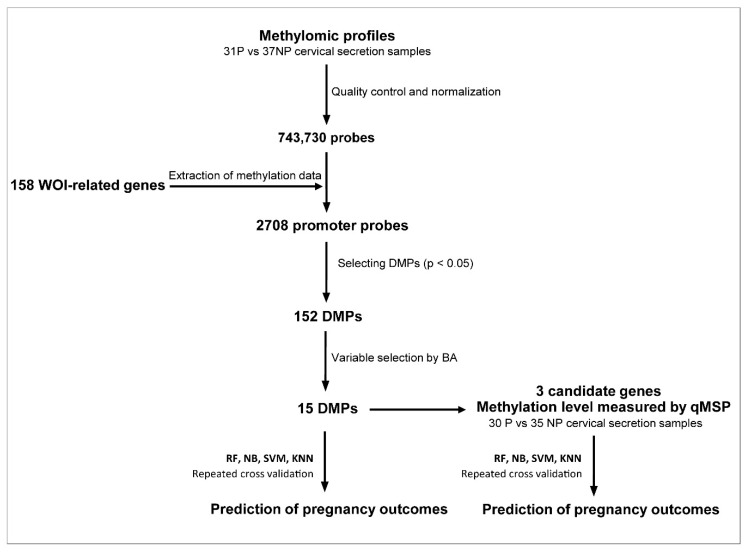
Study logistics. A total of 158 WOI genes were selected from the single cell endometrial transcriptomic profiles. In total, 2708 promoter probes of the 158 WOI genes were extracted from methylomic profiles generated by an EPIC 850K BeadChip array, from 31 pregnancy and 37 nonpregnancy mid-secretory phase cervical secretion samples. Probes were ranked by *p*-value to select 152 DMPs. These DMPs were used in the following refinement step using the Boruta algorithm to identify the 15 DMPs most relevant to ongoing pregnancy status. Three selected genes were verified by qMSP in an independent set of 30 pregnancy and 35 nonpregnancy mid-secretory phase cervical secretion samples. Different sets of selected DMPs or genes were used in four ML algorithms for predicting ongoing pregnancy in the two datasets (array and qMSP datasets). RF: random forest, NB: naïve Bayes, SVM: support vector machine, KNN: k-nearest neighbors, NP: nonpregnancy, P: pregnancy, DMPs: differentially methylated probes, WOI: window of implantation, qMSP: quantitative methylation-specific PCR, CV: cross-validation.

**Figure 2 ijms-24-05598-f002:**
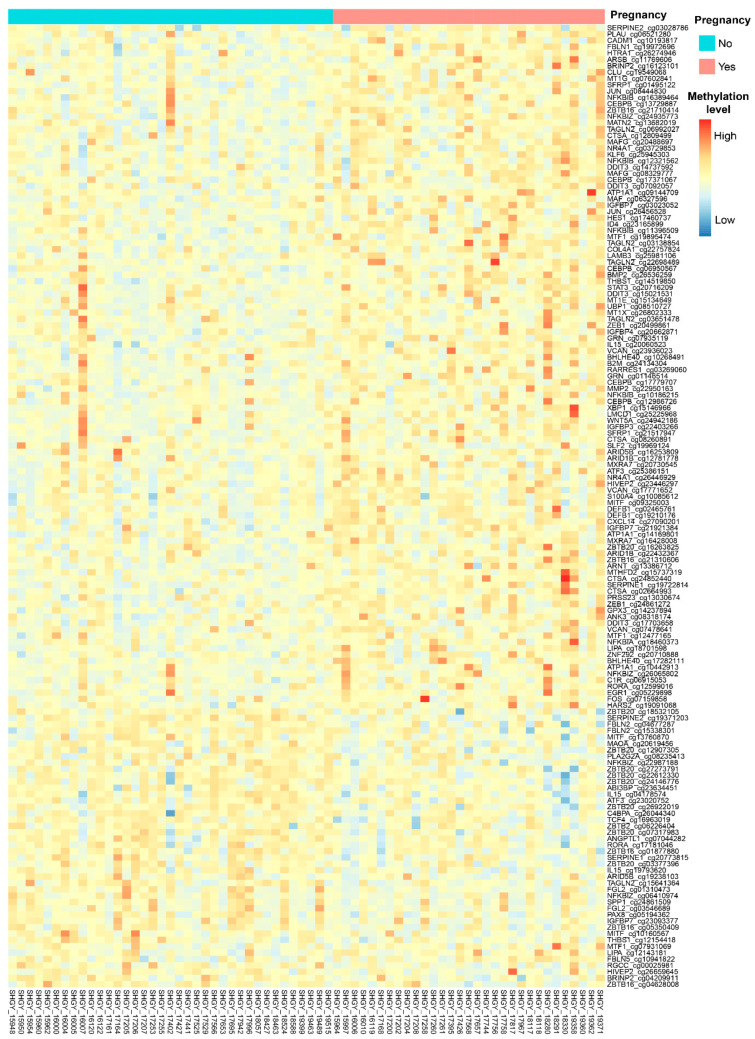
Methylation differences of WOI genes between pregnancy and nonpregnancy cervical secretion samples. A heatmap showing methylation differences of 152 DMPs (87 genes) between the pregnancy and nonpregnancy cervical secretion sample groups. Samples are presented vertically, and values of DNA methylation of the DMPs are presented horizontally. Green and pink columns represent nonpregnancy and pregnancy samples, respectively. Methylation levels are presented as low (dark blue) to high (dark red).

**Figure 3 ijms-24-05598-f003:**
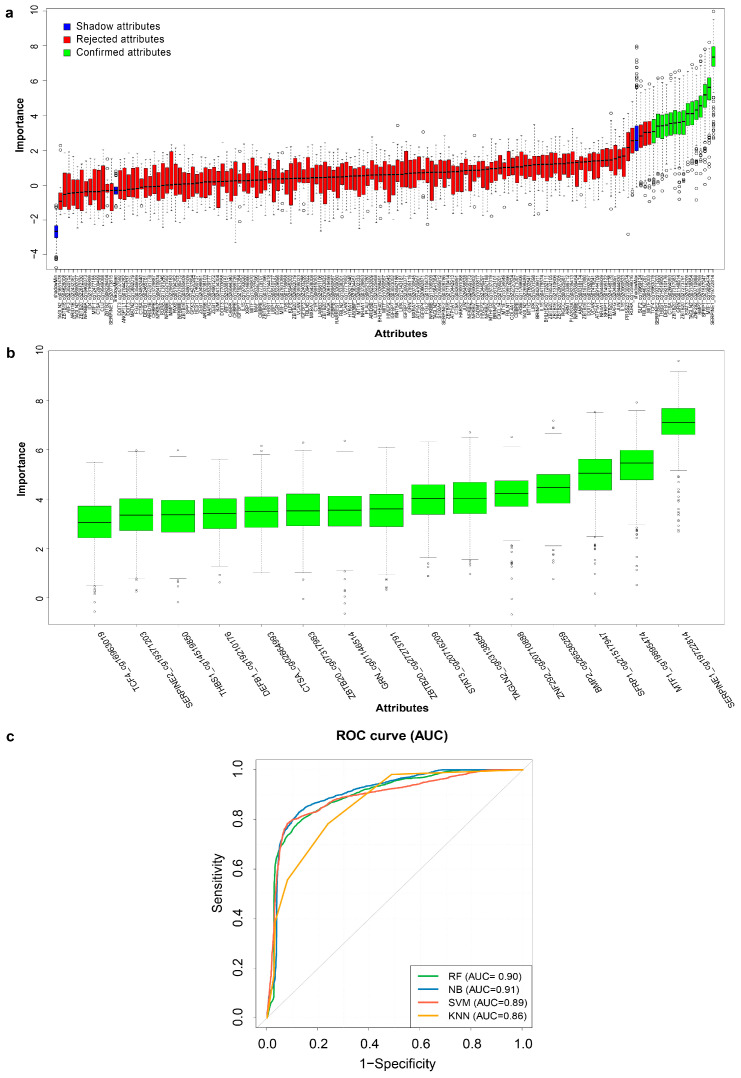
Selection of 15 DMPs and their predictive performance in four ML algorithms. (**a**) A box plot demonstrates the importance ranking of 152 attributes (DMPs) compared with shadow attributes by BA. The blue, red, and green box plots represent the importance levels of the shadow, rejected, and confirmed attributes, respectively. (**b**) Fifteen selected DMPs from the feature selection by BA. (**c**) ROC curves (AUCs) of the best models from four ML algorithms using the 15 selected DMPs for predicting ongoing pregnancy in the array dataset. AUC values were generated from a 100−time repeated fivefold cross-validation. BA: Boruta algorithm, ML: machine learning, DMP: differentially methylated probe, ROC: receiver operating characteristic, AUC: area under the ROC curve, RF: random forest, NB: naïve Bayes, SVM: support vector machine, KNN: k-nearest neighbor.

**Figure 4 ijms-24-05598-f004:**
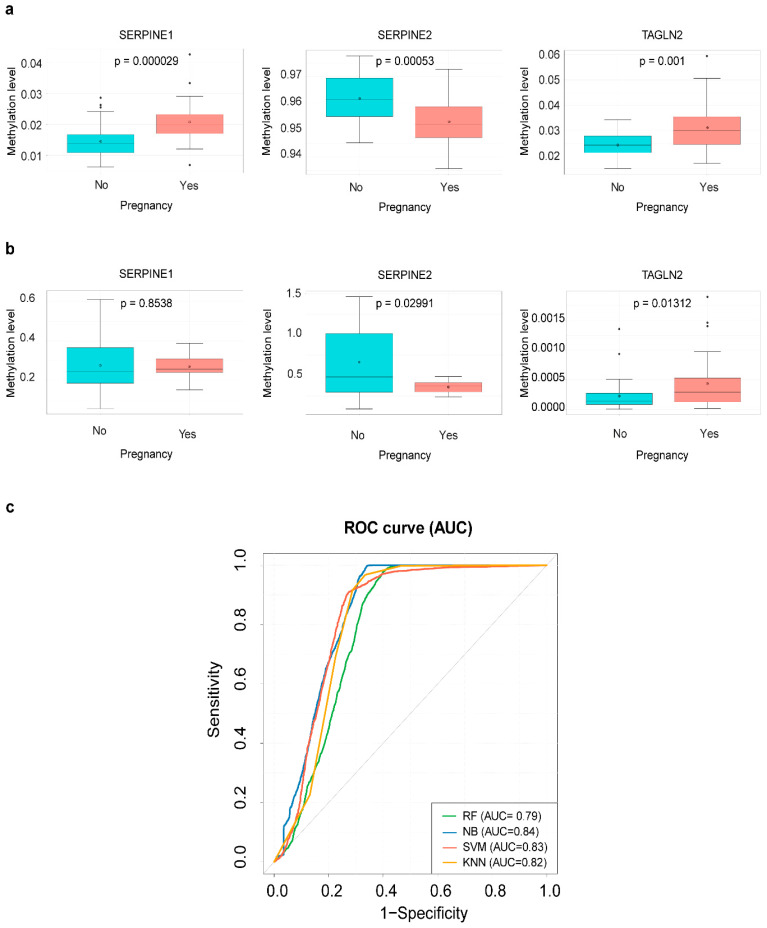
Methylation levels and predictive performance of three selected candidate genes in four ML algorithms. (**a**) A box plot shows differences in methylation levels (β-values) measured by an EPIC 850 K BeadChip array of three selected DMPs from three candidate genes between pregnancy (n = 31) and nonpregnancy (n = 37) cervical secretion samples. (**b**) A box plot shows differences in methylation levels (ΔCp values, transformed into 2^−(ΔCp)^ values) measured by qMSP of three selected candidate genes between pregnancy (n = 30) and nonpregnancy (n = 35) cervical secretion samples. (**c**) ROC curves (AUCs) of the best models of four ML algorithms using three selected candidate genes for predicting ongoing pregnancy in the qMSP dataset. AUC values were generated from a 100−time repeated fivefold cross-validation. ML: machine learning, ROC: receiver operating characteristic, AUC: area under the ROC curve, RF: random forest, NB: naïve Bayes, SVM: support vector machine, KNN: k-nearest neighbor.

**Table 1 ijms-24-05598-t001:** Clinical characteristics of the study samples.

Clinical Characteristics	Array Set			qMSP Set		
	Ongoing Pregnancy		*p* Value	Ongoing Pregnancy		*p* Value
	No (N = 37)	Yes (N = 31)		No (N = 35)	Yes (N = 30)	
Age (years)	36.8 (3.0)	36.0 (2.0)	0.644 ^a^	40.9 (4.5)	38.0 (8.5)	0.515 ^a^
Endometrial thickness (mm)	10.0 (2.8) (n = 32)	9.4 (3.8) (n = 26)	0.165 ^a^	9.8 (2.55) (n = 21)	10.2 (3.25) (n = 18)	0.530 ^a^
Transfer of good quality embryo						
No	8 (21.6)	2 (6.5)	0.097 ^b^	5 (14.3)	0 (0.0)	0.057 ^b^
Yes	29 (78.4)	29 (93.5)		30 (85.7)	34 (100)	
Embryo stage						
Cleavage	19 (51.4)	9 (29.0)	0.063 ^c^	-	-	-
Blastocyst	18 (48.6)	22 (71.0)		35 (100)	30 (100)	
Endometriosis						
No	30 (81.1)	29 (93.5)	0.166 ^b^	28 (80.0)	23 (76.7)	0.745 ^c^
Yes	7 (18.9)	2 (6.5)		7 (20.0)	7 (23.3)	
COH						
No (Frozen-thawed cycle)	35 (94.6)	27 (87.1)	0.4 ^b^	33 (94.3)	30(100)	0.495 ^b^
Yes (Fresh cycle)	2 (5.4)	4 (12.9)		2 (5.7)	0 (0)	
IVF indicator						
Ovulatory	15 (40.5)	11 (35.5)	0.608 ^c^	6 (17.1)	8 (26.7)	0.161 ^c^
Endometriosis	7 (18.9)	2 (6.5)		7 (20.0)	7 (23.3)	
Male	7 (18.9)	6 (19.4)		3 (8.6)	3 (10.0)	
Tubal	2 (5.4)	3 (9.7)		3 (8.6)	0 (0.0)	
Unexplained	5 (13.5)	7 (22.6)		8 (22.8)	9 (30.0)	
Uterine	1 (2.7)	1 (3.2)		0 (0.0)	2 (6.7)	
POF	0 (0.0)	1 (3.2)		-	-	
Advanced women age	-	-		7(20.0)	1 (3.3)	
Recurrent pregnancy loss	-	-		1(2.9)	0 (0.0)	
Endometrial preparation						
Natural	14 (37.8)	19 (61.3)	0.054 ^c^	6 (17.1)	8 (26.7)	0.352 ^c^
Artificial	23 (62.2)	12 (38.7)		29 (82.9)	22 (73.3)	

Data are median (interquartile range) or n (%). *p* values were calculated by (a) Mann-Whitney U test, (b) Fisher’s exact test, or (c) Pearson chi-squared test. For endometrial thickness, statistics were calculated using 58 samples (32 nonpregnancies and 26 pregnancies) in an array (discovery) set, and 39 samples (21 nonpregnancies and 18 pregnancies) in a qMSP (verification) set. COH: controlled ovarian hyperstimulation; IVF: in vitro fertilization; POF: premature ovarian failure; qMSP: quantitative methylation specific polymerase chain reaction.

**Table 2 ijms-24-05598-t002:** Performance of 15 DMPs in four machine learning algorithms.

Model	Accuracy (%)	PPV (%)	NPV (%)	Sensitivity (%)	Specificity (%)
RF	83.53	86.64	81.48	75.52	90.24
NB	85.26	87.36	83.79	79.13	90.41
SVM	85.78	90.57	82.76	76.81	93.30
KNN	76.44	91.89	70.93	53.00	96.08

Data were generated over all 500 resamples in a 100−time repeated fivefold cross-validation procedure. RF: random forest, NB: naïve Bayes, SVM: support vector machine, KNN: k-nearest neighbor. PPV: Positive predictive value, NPV: Negative predictive value, DMPs: Differentially methylated probes.

**Table 3 ijms-24-05598-t003:** Performance of 3 candidate genes in 4 machine learning algorithms.

Model	Accuracy (%)	PPV (%)	NPV (%)	Sensitivity (%)	Specificity (%)
RF	71.46	68.31	74.38	71.20	71.69
NB	80.06	71.98	92.00	93.00	68.97
SVM	80.72	73.34	90.75	91.50	71.49
KNN	80.68	73.19	90.95	91.73	71.20

Data were generated over all 500 resamples in a 100−time repeated fivefold cross-validation procedure. RF: random forest, NB: naïve Bayes, SVM: support vector machine, KNN: k-nearest neighbor. PPV: Positive predictive value, NPV: Negative predictive value, DMPs: Differentially methylated probes.

## Data Availability

The datasets used and/or analyzed during the current study are available from the corresponding author on reasonable request.

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
