# Peer review of "DNA Methylation of Window of Implantation Genes in Cervical Secretions Predicts Ongoing Pregnancy in Infertility Treatment"

_ijms, 2023, doi:10.3390/ijms24065598_

Round 1

Reviewer 1 Report

The present study shows the importance of DNA methylation in assessing pregnancy outcomes by non-invasive methods. The study is appropriately designed and conducted, and the results are correctly presented. This will help the clinicians/personnel involved in IVF to assess the pregnancy outcome. The authors are suggested to study large sample sizes and DNA methylation of WOI genes during different phases of the menstrual cycle in the future. The term DMP should be explained in the manuscript first before using the acronym. 

Author Response

Response to Reviewer 1 Comments

Point 1: The present study shows the importance of DNA methylation in assessing pregnancy outcomes by non-invasive methods. The study is appropriately designed and conducted, and the results are correctly presented. This will help the clinicians/personnel involved in IVF to assess the pregnancy outcome.

Response 1: Many thanks for your comments.

Point 2: The authors are suggested to study large sample sizes and DNA methylation of WOI genes during different phases of the menstrual cycle in the future.

Response 2: Thank you for your suggestion. We planned to validate these methylation markers using late proliferative phase cervical secretions in our ongoing project.

Point 3: The term DMP should be explained in the manuscript first before using the acronym. 

Response 3: Thanks for your comment. DMP is the accronym of “Differentially Methylated Probe”. We add this full term before its accronym “DMP” in the abstract part of our revised manuscript (page 1, line 28).

Reviewer 2 Report

This is a well written and interesting manuscript looking at the predictive value DNA methylation of WOI genes on pregnancy outcome. This reviewer has not recommended edits and recommends accepting the manuscripts as written. 

Author Response

Response to Reviewer 2 Comment

Comment: This is a well written and interesting manuscript looking at the predictive value DNA methylation of WOI genes on pregnancy outcome. This reviewer has not recommended edits and recommends accepting the manuscripts as written. 

Response: Many thanks for your comment.

Reviewer 3 Report

comment 1

Change all the pregnancy outcome to IVF ET outcome

Pregnancy outcome means either one of the followings: abortion, ectopic pregnancy, premature alive or dead delivery, mature alive or dead delivery, post mature alive or dead delivery.

IVF outcome means either one of the followings: biochemical pregnancy due to successful implantation, clinical pregnancy by presence of gestational sac, ongoing pregnancy by the presence of alive successful intrauterine implanted IVF embryo/ embryos, take home baby by alive birth

Comment 2

add the meaning of initials in first appearance in the article (DMP in line 28, all genes in line 28-29, qMSP in table 1)

Comment 3

line 48 change pregnancy rate to take home baby rate

Comment 4

explain which menstrual  cycle's mid secretory phase eg: mock cycle, IVF ET fresh cycle, one cycle prior to IVF or ET cycle, frozen thawed ET cycle)

Author Response

Response to Reviewer 3 Comments

Point 1: Change all the pregnancy outcome to IVF ET outcome

Pregnancy outcome means either one of the followings: abortion, ectopic pregnancy, premature alive or dead delivery, mature alive or dead delivery, post mature alive or dead delivery.

IVF outcome means either one of the followings: biochemical pregnancy due to successful implantation, clinical pregnancy by presence of gestational sac, ongoing pregnancy by the presence of alive successful intrauterine implanted IVF embryo/ embryos, take home baby by alive birth

Response 1: Thanks for your comments. In our manuscript, “pregnancy outcome” means the success or failure of achieving an ongoing pregnancy after embryo transfer. However, we definitely agree with your explanations on the definitions of different reproductive outcomes. We would like to take your suggestions and change “pregnancy outcome” to “ongoing pregnancy” (line 3, 26, 94, 99, 130, 142, 146, 169, 174, 198, 210, 239, 249, 310, 316, 423, 428, 466, 482), or “pregnancy” (line 486), or “ongoing pregnancy status” (line 31, 127, 162, 251, 262, 430, 451 ) or “pregnancy status” (line 308). In addition, we also use the term “IVF-ET oucome” (line 25, 39, 92, 247, 322) or “embryo transfer outcome” (line 341, 484) whenever it is appropriate in our revised manuscript.

Point 2: add the meaning of initials in first appearance in the article (DMP in line 28, all genes in line 28-29, qMSP in table 1).

Response 2: Thanks for your comments. DMP is the accronym of “Differentially Methylated Probe” and qMSP is the accronym of “quantitative Methylation Specific Polymerase Chain Reaction“. We add these full terms before their accronyms in our revised manuscript, accordingly (line 28, 116). In order to keep the Abstract part short and tidy, we would like to add the complete names of 14 genes in the Abbreviation part of our revised manuscript (page 15, line 498-504).

Point 3: line 48 change pregnancy rate to take home baby rate

Response 3: Thanks for your comments. We acknowledge that “take-home baby” is the most wanted result in infertility treatment. However, in our present study, we aimed to look at the potentials of DNA methylations of WOI genes from mid-secretory phase cervical secretions in predicting ongoing pregnancy in embryo transfer cycles. Thus, we would like to keep the term “pregnancy rate” in the introduction part (page 2, line 49 in the revised manuscript) because it is consistent with the outcome variable (ongoing pregnancy) in this study.

Point 4: explain which menstrual cycle's mid secretory phase eg: mock cycle, IVF ET fresh cycle, one cycle prior to IVF or ET cycle, frozen thawed ET cycle)

Response 4: Thanks for your comment. In this study, cervical secretion samples were collected at the mid-secretory phase of the embryo transfer cycles, regardless of using fresh or frozen embryos. The samples were collected right before inserting a catheter into the uterine cavity during the embryo transfer procedure as described in the manuscript.
